# AdvFLYP: Adversarially Finetune Like You Pretrain for Zero-shot Robustness of CLIP

## Abstract

Pretrained vision-language models (VLMs) like CLIP are shown to be highly susceptible to adversarial perturbations. Adversarial finetuning (AFT) approaches have been proposed to improve the zero-shot adversarial robustness of CLIP on various downstream tasks, based on finetuning the vision encoder on adversarial images generated from a proxy classification dataset, such as TinyImageNet. However, we demonstrate that existing AFT approaches have largely overlooked the important role of the training recipe, particularly the training data and objective. To this end, we propose *Adversarially Finetune Like You Pretrain* (AdvFLYP), which practically retains the training recipe of CLIP's pretraining during AFT. We finetune CLIP based on adversarial images generated from web-scale image-text data with a contrastive loss. Experiments validate the superiority of AdvFLYP on various downstream datasets. For example, AdvFLYP outperforms existing AFT approaches finetuned on TinyImageNet (ImageNet) by 19.1% (3.1%), averaged on 14 downstream datasets. Further analyses show that sufficiently large training data amounts and batch sizes are crucial for the contrastive learning of AdvFLYP. Our code and model checkpoints will be released.

## 1 Introduction

Pre-trained vision-language models (VLMs) have been trained to align images with their descriptive texts over significant amounts of image-text pairs, with CLIP (Radford et al., 2021) and ALIGN (Jia et al., 2021) being notable representatives. These models have exhibited remarkable abilities to perform image classification in a zero-shot manner (Pratt et al., 2023; Saha et al., 2024; Sammani & Deligiannis, 2024). However, recent studies have revealed the alarming vulnerabilities of CLIP to adversarial attacks (Mao et al., 2023; Li et al., 2024; Schlarmann et al., 2024): An imperceptible maliciously manipulated noise added to a test image suffices to substantially reduce the model's recognition accuracy.

To enhance CLIP's robustness to adversarial attacks, recent studies introduce adversarial images generated from a proxy dataset such as TinyImageNet (Le & Yang, 2015) or ImageNet (Deng et al., 2009) into the training set, and finetune the vision encoder (Mao et al., 2023; Wang et al., 2024; Yu et al., 2024; Schlarmann et al., 2024) based on adversarial training (AT) (Madry et al., 2018; Zhang et al., 2019). This practice proves effective in improving the model's adversarial robustness on diverse downstream datasets without further training, which is termed *zero-shot robustness* (Mao et al., 2023). However, these methods incur a significant degradation in the model's generalization on clean data of downstream tasks. We hypothesize that such degradation is largely due to the misaligned training recipe between CLIP's pretraining and the finetuning process of existing AFT methods. Intuitively, there is a fundamental difference between adversarially finetuning CLIP and robustly training a model from scratch. CLIP has been pre-trained over web-scale image-text pairs and learned real-world knowledge, and updating its model weights on a specific domain can already lead to noticeable generalization loss (Radford et al., 2021), which further complicates the analysis of this loss in adversarial finetuning. In this work, we investigate the generalization degradation in the adversarial finetuning of CLIP, and identify two important factors: **(1)** the training data distribution that differs from CLIP's pretraining data. This is evidenced by the following observations: (i) Finetuning CLIP on the clean data of a proxy dataset lowers the accuracy on downstream datasets, and (ii) Finetuning CLIP on adversarial images of a proxy dataset (Mao et al., 2023; Wang et al., 2024; Yu et al., 2024) results in higher accuracy on the clean test set of the proxy dataset than the

original CLIP, which indicates that the model has overfit to the distribution of the proxy dataset, even finetuned with adversarial images; **(2)** the training objective for AFT. Minimizing the cross-entropy loss between the generated adversarial images and their correct labels on a classification dataset effectively aligns multiple images from a class to the same textual class name, which causes the loss of knowledge.

Built upon our analysis, we propose a simple yet effective paradigm termed *Adversarially Finetune Like You Pretrain* (AdvFLYP). The main idea of AdvFLYP is to finetune CLIP with adversarial images while maximally retaining the same training recipe as employed in the pretraining phase. Specifically, to imitate the distribution of CLIP's training data, we randomly sample a certain number of web-scale image-text pairs. During AFT, we employ the same contrastive loss for pretraining CLIP. The difference is that we align adversarial images, rather than clean images, with their corresponding texts. To further alleviate the robustness-generalization trade-off, we propose to impose logit- and feature-level regularization during finetuning, which we show to improve robustness transfer and generalization on clean images, respectively. Through extensive experiments, we show that when finetuned with the same amounts of training data, AdvFLYP outperforms existing AFT methods finetuned on TinyImageNet and ImageNet by an average relative improvement of 19.1% and 3.1%, respectively. To facilitate understanding of this paradigm, we vary the training conditions (*e.g.*, batch size, training data amount) and provide insights into contrastive learning of CLIP in the context of adversarial finetuning. We summarize the contributions of this work as follows:

- We investigate the generalization degradation in existing AFT methods for CLIP, and identify two major sources, which are training data distribution and the training objective.
- We introduce *Adversarially Finetune Like You Pretrain* (AdvFLYP), a simple yet effective paradigm to achieve zero-shot adversarial robustness, which resumes contrastive learning of CLIP by aligning adversarial images with their texts.
- Extensive experiments on 14 downstream datasets show that AdvFLYP outperforms mainstream AFT methods. We also vary the training setting of AdvFLYP and show that sufficiently large training data amounts and batch size for AdvFLYP are crucial for robustness and accuracy.

## 2 RELATED WORK

**Adversarial robustness of neural networks.** Deep neural networks (Krizhevsky et al., 2012) are vulnerable to adversarial attacks (Carlini & Wagner, 2017; Szegedy et al., 2014): an imperceptible pixel-level perturbation added to the test image can mislead a well-trained model to make a wrong prediction. Adversarial attacks (Carlini & Wagner, 2017; Croce & Hein, 2020) and defences (Madry et al., 2018) have been extensively studied. Among defence methods, adversarial training (AT) (Madry et al., 2018; Zhang et al., 2019; Rice et al., 2020) has been established as the *de-facto* standard to train an adversarially robust model. More recent research finetunes a standardly trained model on adversarial samples to enhance its adversarial robustness, instead of training a robust model from scratch (Suzuki et al., 2023).

**Adversarial robustness of vision-language models (VLMs)** has also attracted significant research attention (Zhao et al., 2023). In this paper, we focus on *zero-shot adversarial robustness* of CLIP (Radford et al., 2021). Existing methods are largely based on AT, introducing adversarial images into the training set and adapt the CLIP models. There are two types of categories in this regard: *adversarial finetuning* (AFT) (Mao et al., 2023; Wang et al., 2024; Yu et al., 2024; Schlarmann et al., 2024), which finetunes the vision encoder of CLIP; and *adversarial prompt tuning* (Li et al., 2024; Zhang et al., 2024), which learns tunable prompt at the text encoder side to align with adversarial images. More recently, test-time methods for defending CLIP have started to garnered interests (Wang et al., 2025; Sheng et al., 2025; Tong et al., 2025; Zhang et al., 2025; Xing et al., 2025), which achieves inference-time robustness without the need for training. We focus on *adversarial finetuning* methods in this work, which is still the most effective approach. Mao et al. (2023) first propose to generate adversarial images on ImageNet (Deng et al., 2009) by maximizing the cross-entropy loss *w.r.t.* the ground-truth label, which are then leveraged for finetuning vision encoder $f_\theta$ by minimizing the cross-entropy loss of these adversarial images *w.r.t.* the labels. Subsequent work introduces regularization based on this loss. Wang et al. (2024) impose logit-level regularization terms guided by frozen CLIP models to improve robustness on downstream datasets. Yu et al. (2024)

introduce regularization formulated by aligning text-guided attention of the model with the original CLIP. More recently, Dong et al. (2025) focus on improving the adversarial candidates in adversarial finetuning by forming consecutive vertices and sampling simplices. Aside from supervised AFT, recent work proposes unsupervised AFT (Schlarmann et al., 2024). Gong et al. (2025) employ unsupervised AFT as a novel tool to improve interpretability of visual models. These methods employ a proxy dataset and a training objective that differ from CLIP's pretraining. We propose a novel AFT paradigm, AdvFLYP, which challenges the common practice of current AFT methods.

## 3 METHOD

In this section, we first introduce preliminaries regarding CLIP (Radford et al., 2021) and existing finetuning-based methods to achieve *zero-shot adversarial robustness*, and elaborate on our paradigm, termed *Adversarially Finetune Like You Pretrain* (AdvFLYP).

### 3.1 PRELIMINARIES

CLIP (Radford et al., 2021) is a dual-encoder architecture with a vision encoder $f_\theta(\cdot) \in \mathbb{R}^d$ and a text encoder $g_\phi(\cdot) \in \mathbb{R}^d$, which map an image $x$ and a text $t$ into the same $d$-dimensional latent space, respectively. In the pretraining phase of CLIP, the vision and text encoders are trained over 400 million web-scale image-text pairs via a contrastive loss (Oord et al., 2018), which maximizes the cosine similarity of an image embedding with its corresponding text embedding. In a single batch $\{(x_i, t_i)\}_{i=1}^N$, the contrastive loss is formulated as follows:

$$\mathcal{L}_{CLIP}\left(\{(x_i, t_i)\}_{i=1}^N\right) = -\frac{1}{2N} \sum_{i=1}^N \left[ \log \frac{\exp(s_{ii}/\tau)}{\sum_{j=1}^N \exp(s_{ij}/\tau)} + \log \frac{\exp(s_{ii}/\tau)}{\sum_{j=1}^N \exp(s_{ji}/\tau)} \right] \quad (1)$$

where $\tau$ is the temperature, and $s_{ij} = \frac{f_\theta(x_i)^\mathsf{T} g_\phi(t_j)}{\|f_\theta(x_i)\|\|g_\phi(t_j)\|}$ is the cosine similarity between $x_i$ and $t_j$. After the pretraining phase, given an image $x_{test}$ and a set of pre-trained textual categories $\{c_1, \ldots c_K\}$ at inference time, CLIP is able to perform zero-shot classification by classifying it as the category with the highest similarity $\hat{y} = \arg\max_k \frac{f_\theta(x_{test})^\mathsf{T} g_\phi(T[c_k])}{\|f_\theta(x_{test})\| \cdot \|g_\phi(T[c_k])\|}$, where $T[\cdot]$ is a textual template, which is usually *'a photo of a [CLS]'*.

**Adversarial attacks**. A pixel-level perturbation $\delta \in \mathbb{R}^{C \times H \times W}$ bounded by a $L_\infty$-radius ball, when maliciously designed to maximize the loss of a given image $x$ *w.r.t.* its label $c_{GT}$, can cause CLIP to misclassify the sample:

$$\delta_{adv} = \arg\max_\delta \mathcal{L}\left(f_\theta(x + \delta), c_{GT}\right), \quad s.t. \|\delta\|_\infty \leq \epsilon \quad (2)$$

where $\mathcal{L}$ is cross-entropy loss, and $\epsilon$ is the attack budget controlling the attack strength.

**Adversarial finetuning of CLIP** typically finetunes the pre-trained vision encoder $f_\theta$ by generating adversarial images on the fly and aligning them with their correct labels on a proxy dataset. To this end, Mao et al. (2023) propose TeCoA, which is a conventional cross-entropy loss of adversarial images w.r.t. ground-truth labels:

$$\theta' = \arg\min_\theta \mathcal{L}\left(f_\theta(x + \delta_{adv}), c_{GT}\right) \quad (3)$$

Subsequent finetuning-based methods (Wang et al., 2024; Yu et al., 2024) introduce regularization terms to improve robustness on downstream datasets and generalization on top of this loss. Specifically, Wang et al. (2024) employ a frozen original CLIP model to guide the finetuning process, while Yu et al. (2024) propose text-guided attention and regularize the finetuning and encourage the model to attend to informative areas in adversarial images.

### 3.2 LIMITATIONS OF EXISTING METHODS

Despite their effectiveness in boosting zero-shot adversarial robustness, these methods incur a significant generalization decline on clean data (Mao et al., 2023). To look into this degradation, we perform introductory experiments by leveraging two proxy datasets, ImageNet (Deng et al., 2009)

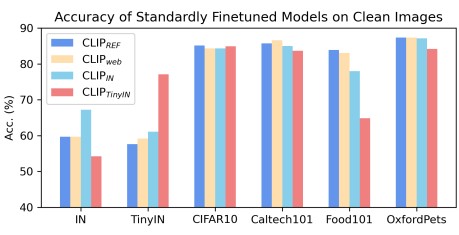 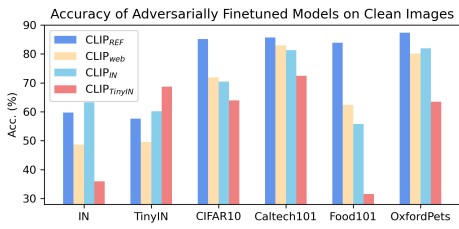

(a) standardly finetuned models        (b) adversarially finetuned models

Figure 1: Behaviour of finetuned models. The subscripts in the legends indicate the dataset used for finetuning. $\text{CLIP}_{web}$ represents CLIP finetuned on a toy dataset of 100k web-scale image-text pairs. $\text{CLIP}_{REF}$ denotes the original pre-trained CLIP without any weight updates.

and TinyImageNet (Le & Yang, 2015), to finetune $f_\theta$ of CLIP with either clean images (a.k.a. standard finetuning) or adversarial images (a.k.a. adversarial finetuning). We also collect a toy web-scale dataset of 100k image-text pairs to imitate CLIP's pre-training data distribution, denoted as *web*. We test these models on the clean images of 6 selected datasets, which include two proxy datasets involved, two general object recognition datasets CIFAR10 (Krizhevsky et al., 2009) and Caltech101 (Fei-Fei et al., 2006), and two fine-grained datasets Food101 (Bossard et al., 2014) and OxfordPets (Parkhi et al., 2012). We report the results in Fig.1 and make the following observations: (1) The model still suffers a noticeable loss in generalization ability even when it is finetuned with clean images on a proxy dataset. (2) Adversarial finetuning of CLIP on a proxy dataset leads to higher accuracy on clean images from this dataset than the original CLIP. Both observations indicate that apart from the inherent robustness-generalization trade-off, the finetuned CLIP overfits to the data distribution of the dataset it has been finetuned on, therefore compromising generalization further.

Furthermore, when finetuning $f_\theta$ on a classification dataset via a cross-entropy loss, it equivalently aligns a large number of semantically-rich images from the same category with a single textual prompt. Intuitively, this forces the model to adapt to the classification task instead of retaining its capability of matching images and texts.

### 3.3 ADVFLYP

To address the limitations discussed above, we propose a simple yet effective adversarial finetuning paradigm, which we term *Adversarially Finetune Like You Pretrain* (AdvFLYP)[1], to achieve *zero-shot adversarial robustness*. The idea is intuitive, and can be viewed as resuming the training of CLIP with adversarial images while maximally maintaining the training recipe.

**Data preparation.** Since the pretraining data of CLIP is not publicly available, to imitate the distribution of CLIP's pre-training data, we collect 1M webscale image-text pairs. To this end, we randomly sample one million entries with reachable URLs

---

**Algorithm 1** PyTorch-style pseudocode for AdvFLYP

```python
# target vision encoder f_theta
# frozen orignal vision encoder F_theta0
# frozen text encoder g_phi
# collected data: web image-text pairs D
for (X, T) in D: # one batch
    # generate adversarial perturbations
    delta=PGD(f, g, (X, T), l_clip)
    # obtain embeddings
    X=f(X+delta),X_c=f(X),X_ori=F(X+delta),T=g(T)
    # compute probability logit
    P=X@T.t(), P_c=X_c@T.t(), P_ori=X_ori@T.t()
    # logit-level regularization
    l_logit=P*(P/P_c).log()+P*(P/P_ori).log()
    # feature-level regularization
    l_feat=(X-X_c).norm(-1)+(X-X_ori).norm(-1)
    # update theta w.r.t. final loss
    L=(l_clip(f,g,(X_B+delta,T_B))+l_logit+l_feat)
    L.backward()
    optimizer.step()
return theta
```

---

from LAION-400M (Schuhmann et al., 2021). Following the original work of CLIP (Radford et al., 2021), we utilize these noisy web-scale data without further data cleansing.

---

[1]This paradigm is named after a work on robust finetuning of CLIP, *Finetune Like You Pretrain* (FLYP) (Goyal et al., 2023), which finds that CLIP fintuned with the same contrastive objective as in pretraining compares favourably to CLIP typically finetuned with a cross-entropy loss.

**Adversarial finetuning.** As in general adversarial training (AT) frameworks (Madry et al., 2018), this process involves a min-max optimization. Instead of employing a cross-entropy loss on adversarial images from a classification dataset (Mao et al., 2023; Wang et al., 2024; Yu et al., 2024), we propose to employ the same contrastive loss as in CLIP's pretraining (Eq. 1) in our adversarial finetuning paradigm. Specifically, given a batch of image-text data $\{(x_i, t_i)\}_{i=1}^N$, in the inner maximization process, we optimize a $L_\infty$-bounded perturbation $\delta_i$ for each sample $x_i$ in this batch, such that this contrastive loss (Eq. 1) is maximized:

$$\boldsymbol{\delta} = \arg \max_{\{\delta_1,\dots,\delta_N\}} \mathcal{L}_{CLIP}\left(\{(x_i + \delta_i, t_i)\}_{i=1}^N\right), \quad s.t. \|\delta\|_\infty \leq \epsilon \tag{4}$$

Note that $\boldsymbol{\delta} \in \mathbb{R}^{N \times C \times H \times W}$ is optimized at the same time, instead of being optimized individually, by employing PGD algorithm (Carlini & Wagner, 2017). This is in stark contrast to existing adversarial finetuning methods (Mao et al., 2023), where a perturbation is optimized independently for each image to maximize its cross-entropy loss against a pre-defined set of categories (Eq. 2). In the experiment section, we will investigate the impact of the contrastive learning setting, *e.g.*, batch size, in the context of adversarial finetuning. In the outer minimization loop, we finetune the model weights $\theta$ of the vision encoder to minimize the contrastive loss of this batch of adversarial samples. To further alleviate generalization loss, we also incorporate regularization guided by the frozen original CLIP $F_{\theta_0}$ during funetuning. Specifically, we obtain the normalized embeddings of adversarial images, $X_\theta^{adv} = \left[\frac{f_\theta(x_i + \delta_i)}{\|f_\theta(x_i + \delta_i)\|}\right]_{i=1}^N \in \mathbb{R}^{N \times d}$ and $X_{\theta_0}^{adv} = \left[\frac{f_{\theta_0}(x_i + \delta_i)}{\|f_{\theta_0}(x_i + \delta_i)\|}\right]_{i=1}^N \in \mathbb{R}^{N \times d}$, which are output by the target model $f_\theta$ and the original model $F_\theta$, respectively. We also feed the clean images to the target model and obtain their embeddings $X_\theta^{clean} = \left[\frac{f_\theta(x_i)}{\|f_\theta(x_i)\|}\right]_{i=1}^N \in \mathbb{R}^{N \times d}$, We compute the probability logits of $X_\theta^{adv}$, $X_{\theta_0}^{adv}$ and $X_\theta^{clean}$ *w.r.t.* the text features $T_\phi = \left[\frac{g_\phi(t_i)}{\|g_\phi(t_i)\|}\right]_{i=1}^N \in \mathbb{R}^{N \times d}$:

$$P_\theta^{adv} = \text{softmax}(X_\theta^{adv} T^\intercal) \in \mathbb{R}^{N \times N} \tag{5}$$

$$P_{\theta_0}^{adv} = \text{softmax}(X_{\theta_0}^{adv} T^\intercal) \in \mathbb{R}^{N \times N} \tag{6}$$

$$P_\theta^{clean} = \text{softmax}(X_\theta^{clean} T^\intercal) \in \mathbb{R}^{N \times N} \tag{7}$$

The logit-level regularization is formulated following Wang et al. (2024):

$$\mathcal{L}_{logit} = \frac{1}{N}\left[\text{KL}(P_\theta^{adv} \| P_{\theta_0}^{adv}) + \text{KL}(P_\theta^{adv} \| P_\theta^{clean})\right] \tag{8}$$

where $\text{KL}(\cdot\|\cdot)$ denotes KL divergence. In this work, we additionally introduce feature-level regularization, which we find to benefit generalization on clean images:

$$\mathcal{L}_{feat} = \frac{1}{N}\left[\|X_\theta^{adv} - X_{\theta_0}^{adv}\|_F + \|X_\theta^{adv} - X_\theta^{clean}\|_F\right] \tag{9}$$

where $\|\cdot\|_F$ denotes Frobenius norm. To sum up, in the outer minimization loop, the weights of the vision encoder $f_\theta$ are updated as follows:

$$\theta' = \arg \min_\theta \{\mathcal{L}_{CLIP}\left(\{(x_i + \delta_i, t_i)\}_{i=1}^N\right) + \mathcal{L}_{logit} + \mathcal{L}_{feat}\} \tag{10}$$

We summarize the paradigm of AdvFLYP in Alg. 1.

## 4 EXPERIMENTS

We conduct extensive experiments to evaluate the adversarial robustness of our proposed paradigm. We implement state-of-the-art finetuning-based methods with available code on two common proxy datasets, TinyImageNet and ImageNet, and compare AdvFLYP with these baselines under various attack scenarios. We also implement AdvFLYP under multiple finetuning settings to understand the behaviour of this contrastive learning framework in an adversarial finetuning (AFT) context.

### 4.1 Implementation Details

Following previous AFT-based methods, we finetune the pre-trained CLIP's ViT-B/32 vision encoder. To prepare web-scale image text-pairs that closely follow CLIP's pre-training data distribution, we collect 1M data pairs. Specifically, we randomly sample one million data points with reachable URLs from LAION-400M (Schuhmann et al., 2021) and denote it as `small-LAION`. Following the data preprocessing of CLIP, we crop and resize the raw images to the size of $224 \times 224$. In the finetuning process, we set the batch size to 256, unless otherwise specified. To generate adversarial images, we employ the PGD algorithm (Carlini & Wagner, 2017) with 2 iterations to update the batch-wise perturbations $\delta \in \mathbb{R}^{N \times C \times H \times W}$ (Eq. 4). The attack strength and step size during finetuning are set to $\epsilon = 1/255$, $\alpha = 1/255$, respectively. We leverage an SGD optimizer and retain the initial learning rate from CLIP's pre-training stage at $1e-4$ (Radford et al., 2021), which is dynamically adjusted with cosine scheduling. We finetune the model for 20 epochs on a single NVIDIA RTX A6000 GPU device.

### 4.2 Baselines and Datasets

We implement TeCoA (Mao et al., 2023), PMG-AFT (Wang et al., 2024) and TGA-ZSR (Yu et al., 2024) based on their released code and finetune $f_\theta$ on two proxy datasets, TinyImageNet (Le & Yang, 2015) and ImageNet (Deng et al., 2009), which include 100k and roughly 1.2M training images, respectively. The main training objective of these methods is the cross-entropy loss of adversarial images *w.r.t.* their true labels on a classification dataset. To ensure fair comparison, we use their original hyperparameters in our implementation while keeping other settings such as dataset pre-processing strictly identical. We further implement FARE (Schlarmann et al., 2024), which is an unsupervised adversarial finetuning method that alternately generates adversarial images by enlarging their the $L_2$ distance to the original embeddings in the latent space, and updates the encoder weights to minimize their distance. When comparing to baselines finetuned on TinyImageNet, we randomly sample a subset of 100k training image-text pairs from `small-LAION` to maintain the same training data amount, which we denote as `tiny-LAION`.

After the finetuning process, we evaluate the *zero-shot adversarial robustness* of all baselines on 14 downstream datasets spanning diverse domains, which include general object recognition datasets CIFAR10 (Krizhevsky et al., 2009), CIFAR100 (Krizhevsky et al., 2009), STL10 (Coates et al., 2011), Caltech101 (Fei-Fei et al., 2006) and Caltech256 (Griffin et al., 2007); fine-grained recognition datasets OxfordPets (Parkhi et al., 2012), Flowers102 (Nilsback & Zisserman, 2008), Food101 (Bossard et al., 2014), StanfordCars (Krause et al., 2013); scene recognition datasets SUN397 (Xiao et al., 2010) and Country211 (Radford et al., 2021); domain-specific datasets FGVCAircraft (Maji et al., 2013), EuroSAT (Helber et al., 2019), DTD (Cimpoi et al., 2014).

### 4.3 Results and Discussion

**AdvFLYP *v.s.* AFT methods finetuned on TinyIN.** Although recent work (Wang et al., 2024; Yu et al., 2024) employs TinyImageNet as a common proxy dataset due to its small size, we argue that it is not an ideal dataset for AFT. In our preliminary experiments (Fig. 1a in Sec. 3.2), we find that when performing standard finetuning on TinyImageNet, it already causes a significant accuracy degradation on downstream tasks. As can be seen in Table 1, in AFT, this degradation is further worsened by introducing adversarial images, with TeCoA and PMG-AFT losing over 20 average points on downstream datasets compared to the original CLIP. This degradation is largely attributed to the fact that TinyImageNet, with very limited semantics in low-resolution ($64 \times 64$) training images, drastically differs from the distribution of CLIP's pre-training data, which are noisy web-scale image-text pairs. All AFT baselines exhibit substantially higher accuracy of clean images on TinyImageNet, which indicates that they heavily overfit to the data distribution of TinyImageNet, even when finetuned with adversarial images. Despite weaker zero-shot adversarial robustness compared to supervised AFT methods, FARE retains the best clean accuracy among other baselines, showing that unsupervised AFT better preserves CLIP's zero-shot capabilities than supervised counterparts. AdvFLYP, finetuned with a contrastive loss on the same amount of images collected from the web, achieves the same level of clean accuracy with unsupervised AFT, while showing higher adversarial robustness than supervised AFT methods under PGD-10 (Carlini & Wagner, 2017) and AutoAttack (Croce & Hein, 2020) at attack strength $\epsilon = 1/255$. When evaluated under PGD-10 at $\epsilon = 4/255$,

| (%) | | TinyIN | CIFAR10 | CIFAR100 | STL10 | Caltech101 | Caltech256 | OxfordPets | Flowers102 | Food101 | StanfordCars | SUN397 | Country211 | FGVCAircraft | EuroSAT | DTD | avg. |
|---|---|---|---|---|---|---|---|---|---|---|---|---|---|---|---|---|---|
| PGD-10 ($\epsilon = 1/255$) | CLIP | 0.21 | 0.64 | 0.18 | 11.38 | 14.82 | 8.38 | 1.09 | 1.04 | 0.67 | 0.03 | 1.15 | 0.03 | 0.00 | 0.01 | 3.09 | 3.04 |
| | FARE | 18.36 | 17.85 | 9.77 | 56.30 | 51.26 | 37.10 | 29.22 | 15.14 | 10.67 | 6.84 | 13.63 | 0.78 | 1.08 | 9.64 | 14.63 | 19.56 |
| | TeCoA | 45.34 | 31.29 | 17.88 | 69.06 | 55.63 | 43.14 | 38.35 | 22.28 | 14.30 | 8.89 | 19.94 | 1.84 | 2.25 | 11.55 | 17.50 | 25.28 |
| | PMG-AFT | 46.13 | 40.68 | 22.53 | 73.09 | 61.12 | 45.92 | 41.18 | 23.50 | 18.57 | 11.65 | 22.58 | 2.10 | 2.19 | 12.60 | 15.00 | 28.05 |
| | TGA-ZSR | 50.23 | 38.49 | 21.46 | 71.91 | 59.38 | 48.81 | 42.63 | 27.32 | 17.78 | 12.00 | 22.39 | 1.92 | 3.93 | 11.63 | 18.99 | 28.47 |
| | AdvFLYP | 23.57 | 39.06 | 18.00 | 71.65 | 67.80 | 56.49 | 53.37 | 31.16 | 29.89 | 19.20 | 29.46 | 3.18 | 5.76 | 15.50 | 22.29 | 33.06 |
| PGD-10 ($\epsilon = 4/255$) | CLIP | 0.00 | 0.00 | 0.00 | 0.01 | 0.59 | 0.14 | 0.00 | 0.00 | 0.00 | 0.00 | 0.00 | 0.00 | 0.00 | 0.00 | 0.11 | 0.06 |
| | FARE | 0.22 | 0.01 | 0.04 | 1.00 | 4.77 | 1.96 | 0.27 | 0.00 | 0.04 | 0.01 | 0.07 | 0.01 | 0.00 | 0.00 | 0.64 | 0.63 |
| | TeCoA | 3.31 | 0.38 | 1.39 | 10.81 | 14.44 | 8.01 | 0.76 | 1.69 | 0.54 | 0.10 | 1.19 | 0.04 | 0.03 | 9.86 | 4.20 | 3.82 |
| | PMG-AFT | 4.44 | 1.21 | 1.72 | 15.06 | 19.47 | 10.63 | 1.72 | 2.50 | 1.03 | 0.12 | 1.89 | 0.12 | 0.03 | 9.62 | 4.31 | 4.96 |
| | TGA-ZSR | 2.31 | 0.09 | 0.34 | 6.34 | 11.19 | 5.70 | 0.27 | 0.62 | 0.16 | 0.03 | 0.62 | 0.02 | 0.03 | 0.02 | 2.23 | 1.98 |
| | AdvFLYP | 0.57 | 0.30 | 0.74 | 9.25 | 17.85 | 10.14 | 0.71 | 1.37 | 0.45 | 0.11 | 1.21 | 0.02 | 0.00 | 2.45 | 3.72 | 3.45 |
| AutoAttack ($\epsilon = 1/255$) | CLIP | 0.05 | 0.00 | 0.01 | 0.00 | 0.44 | 0.10 | 0.00 | 0.03 | 0.00 | 0.11 | 0.01 | 0.01 | 0.18 | 0.06 | 0.21 | 0.08 |
| | FARE | 16.44 | 15.29 | 8.27 | 54.38 | 48.83 | 34.88 | 26.68 | 13.30 | 9.59 | 5.36 | 11.23 | 0.48 | 0.72 | 7.74 | 11.70 | 17.75 |
| | TeCoA | 42.89 | 29.71 | 16.49 | 68.30 | 54.48 | 41.75 | 37.12 | 20.78 | 12.63 | 7.70 | 18.07 | 1.43 | 1.53 | 11.29 | 16.54 | 24.13 |
| | PMG-AFT | 43.19 | 38.42 | 20.20 | 72.25 | 59.53 | 44.18 | 38.73 | 21.29 | 16.27 | 9.45 | 20.12 | 1.73 | 1.68 | 11.92 | 13.78 | 26.40 |
| | TGA-ZSR | 29.76 | 17.22 | 10.84 | 56.29 | 47.36 | 36.95 | 28.37 | 16.96 | 10.69 | 5.63 | 11.82 | 0.68 | 1.47 | 8.97 | 12.39 | 18.97 |
| | AdvFLYP | 21.03 | 35.26 | 15.90 | 70.99 | 66.97 | 55.12 | 51.54 | 29.13 | 28.17 | 16.67 | 27.26 | 2.48 | 4.20 | 13.38 | 20.16 | 31.23 |
| clean | CLIP | 57.64 | 85.09 | 57.13 | 96.41 | 85.70 | 81.74 | 87.33 | 65.46 | 83.88 | 52.07 | 58.51 | 15.22 | 20.16 | 42.53 | 40.48 | 62.26 |
| | FARE | 67.82 | 79.09 | 49.63 | 92.05 | 84.14 | 75.00 | 80.35 | 49.31 | 59.09 | 43.48 | 53.38 | 10.10 | 11.94 | 28.35 | 33.99 | 53.56 |
| | TeCoA | 68.65 | 63.95 | 35.27 | 87.20 | 72.44 | 61.77 | 63.45 | 37.63 | 31.57 | 22.30 | 38.08 | 5.14 | 5.79 | 14.62 | 25.85 | 40.36 |
| | PMG-AFT | 66.80 | 70.66 | 40.29 | 88.60 | 75.48 | 62.26 | 65.88 | 37.03 | 36.63 | 25.40 | 37.96 | 4.64 | 5.46 | 18.51 | 21.92 | 42.19 |
| | TGA-ZSR | 76.14 | 81.99 | 53.80 | 90.80 | 79.07 | 72.54 | 74.22 | 46.76 | 49.75 | 34.67 | 48.20 | 7.76 | 11.31 | 22.86 | 30.37 | 50.29 |
| | AdvFLYP | 49.14 | 71.52 | 37.12 | 89.99 | 83.39 | 76.33 | 81.52 | 53.29 | 64.50 | 45.28 | 54.46 | 9.81 | 16.17 | 27.33 | 35.21 | 53.28 |
| $\overline{AVG}$ | CLIP | 14.48 | 21.43 | 14.33 | 26.95 | 25.39 | 22.59 | 22.11 | 16.63 | 21.14 | 13.05 | 14.92 | 3.82 | 5.09 | 10.65 | 10.97 | 16.36 |
| | FARE | 25.71 | 28.06 | 16.93 | 50.93 | 47.25 | 37.25 | 34.13 | 19.44 | 19.85 | 13.92 | 19.58 | 2.84 | 3.44 | 11.43 | 15.24 | 22.88 |
| | TeCoA | 40.05 | 31.33 | 17.76 | 58.84 | 49.25 | 38.67 | 34.92 | 20.59 | 14.76 | 9.75 | 19.32 | 2.11 | 2.40 | 11.83 | 16.02 | 23.40 |
| | PMG-AFT | 40.14 | 37.74 | 21.19 | 62.25 | 53.90 | 40.75 | 36.88 | 21.08 | 18.12 | 11.65 | 20.64 | 2.15 | 2.34 | 13.16 | 13.75 | 25.40 |
| | TGA-ZSR | 39.61 | 34.45 | 21.61 | 56.33 | 49.25 | 41.00 | 36.37 | 22.91 | 19.59 | 13.08 | 20.76 | 2.59 | 4.19 | 10.87 | 16.00 | 24.93 |
| | AdvFLYP | 23.58 | 36.64 | 17.94 | 60.47 | 59.00 | 49.52 | 46.78 | 28.74 | 30.75 | 20.32 | 28.10 | 3.87 | 6.53 | 14.66 | 20.34 | 30.26 |

Table 1: Recognition accuracy under different attack scenarios ($\epsilon = 1/255, 4/255$ and clean images) on downstream datasets of CLIP finetuned with AdvFLYP on `tiny-LAION`, compared with existing methods finetuned on TinyImageNet. We highlight the **best** and second best results.

| (%) | | ImageNet | CIFAR10 | CIFAR100 | STL10 | Caltech101 | Caltech256 | OxfordPets | Flowers102 | Food101 | StanfordCars | SUN397 | Country211 | FGVCAircraft | EuroSAT | DTD | avg. |
|---|---|---|---|---|---|---|---|---|---|---|---|---|---|---|---|---|---|
| PGD-10 ($\epsilon = 1/255$) | CLIP | 0.21 | 0.64 | 0.18 | 11.38 | 14.82 | 8.38 | 1.09 | 1.04 | 0.67 | 0.03 | 1.15 | 0.03 | 0.00 | 0.01 | 3.09 | 3.04 |
| | FARE | 22.36 | 24.79 | 11.50 | 63.80 | 59.46 | 49.21 | 45.71 | 22.20 | 19.80 | 8.97 | 19.31 | 1.07 | 2.76 | 5.96 | 19.73 | 25.30 |
| | TeCoA | 40.83 | 37.31 | 19.64 | 75.49 | 69.21 | 59.47 | 61.35 | 31.13 | 27.87 | 13.29 | 31.17 | 3.22 | 5.55 | 15.13 | 22.34 | 33.73 |
| | PMG-AFT | 39.43 | 42.19 | 21.78 | 77.34 | 72.00 | 61.06 | 64.21 | 33.94 | 33.11 | 16.99 | 32.21 | 3.25 | 6.03 | 14.87 | 14.05 | 35.97 |
| | TGA-ZSR | 56.52 | 37.80 | 19.15 | 79.33 | 75.53 | 64.76 | 73.32 | 34.07 | 37.32 | 19.05 | 37.28 | 3.49 | 8.67 | 14.05 | 26.17 | 37.86 |
| | AdvFLYP | 29.58 | 49.36 | 24.31 | 76.59 | 71.40 | 60.90 | 58.79 | 37.03 | 36.54 | 23.95 | 33.87 | 3.88 | 7.02 | 12.53 | 24.84 | 37.22 |
| PGD-10 ($\epsilon = 4/255$) | CLIP | 0.00 | 0.00 | 0.00 | 0.01 | 0.59 | 0.14 | 0.00 | 0.00 | 0.00 | 0.00 | 0.00 | 0.00 | 0.00 | 0.00 | 0.11 | 0.06 |
| | FARE | 0.28 | 0.02 | 0.00 | 1.39 | 6.57 | 2.92 | 1.55 | 0.00 | 0.03 | 0.00 | 0.10 | 0.00 | 0.00 | 0.00 | 0.85 | 0.96 |
| | TeCoA | 3.54 | 0.70 | 0.76 | 9.31 | 20.41 | 12.42 | 3.22 | 2.02 | 0.63 | 0.09 | 2.06 | 0.11 | 0.00 | 5.60 | 4.68 | 4.43 |
| | PMG-AFT | 3.32 | 0.52 | 0.97 | 10.01 | 20.04 | 12.32 | 2.89 | 1.76 | 0.80 | 0.08 | 1.95 | 0.07 | 0.03 | 6.52 | 4.84 | 4.48 |
| | TGA-ZSR | 0.20 | 0.01 | 0.00 | 2.03 | 6.70 | 3.35 | 2.02 | 0.02 | 0.11 | 0.01 | 0.26 | 0.01 | 0.00 | 0.00 | 0.48 | 1.07 |
| | AdvFLYP | 2.05 | 1.42 | 1.99 | 15.81 | 23.13 | 14.22 | 2.10 | 2.07 | 0.94 | 0.26 | 2.30 | 0.07 | 0.00 | 0.06 | 5.75 | 5.01 |
| AutoAttack ($\epsilon = 1/255$) | CLIP | 0.05 | 0.00 | 0.01 | 0.00 | 0.44 | 0.10 | 0.00 | 0.03 | 0.00 | 0.11 | 0.01 | 0.01 | 0.18 | 0.06 | 0.21 | 0.08 |
| | FARE | 21.46 | 23.07 | 10.59 | 62.94 | 58.59 | 48.29 | 44.89 | 21.03 | 19.02 | 7.90 | 17.73 | 0.86 | 2.01 | 5.36 | 18.14 | 24.32 |
| | TeCoA | 38.58 | 35.39 | 17.92 | 74.95 | 68.54 | 58.23 | 60.13 | 29.06 | 25.78 | 11.69 | 28.77 | 2.66 | 4.47 | 12.98 | 20.75 | 32.21 |
| | PMG-AFT | 36.90 | 39.57 | 19.67 | 76.76 | 71.29 | 59.60 | 62.44 | 31.09 | 30.38 | 14.59 | 29.56 | 2.65 | 4.83 | 13.01 | 21.81 | 34.09 |
| | TGA-ZSR | 0.06 | 0.04 | 0.02 | 0.08 | 0.35 | 0.11 | 0.08 | 0.05 | 0.02 | 0.00 | 0.01 | 0.03 | 0.12 | 0.08 | 0.16 | 0.08 |
| | AdvFLYP | 27.72 | 46.55 | 21.90 | 75.98 | 70.67 | 59.73 | 57.56 | 34.79 | 34.59 | 21.38 | 31.61 | 3.13 | 5.28 | 11.16 | 23.19 | 35.54 |
| clean | CLIP | 57.64 | 85.09 | 57.13 | 96.41 | 85.70 | 81.74 | 87.33 | 65.46 | 83.88 | 52.07 | 58.51 | 15.22 | 20.16 | 42.53 | 40.48 | 62.26 |
| | FARE | 61.80 | 79.25 | 53.22 | 94.46 | 86.25 | 81.63 | 87.33 | 62.48 | 74.59 | 49.45 | 59.57 | 11.80 | 19.89 | 26.77 | 39.47 | 59.01 |
| | TeCoA | 63.28 | 70.40 | 39.82 | 91.45 | 81.31 | 76.72 | 81.98 | 51.65 | 55.78 | 34.08 | 53.29 | 8.11 | 13.35 | 26.72 | 33.83 | 51.32 |
| | PMG-AFT | 61.82 | 75.26 | 43.88 | 92.81 | 84.65 | 78.39 | 83.92 | 55.34 | 64.70 | 42.08 | 55.33 | 9.25 | 14.85 | 24.67 | 34.57 | 54.27 |
| | TGA-ZSR | 68.23 | 77.23 | 41.82 | 91.59 | 77.88 | 77.56 | 81.22 | 52.85 | 66.33 | 37.72 | 53.09 | 9.85 | 12.36 | 32.84 | 35.53 | 53.42 |
| | AdvFLYP | 51.63 | 75.39 | 44.02 | 91.34 | 84.26 | 77.53 | 83.10 | 56.33 | 68.00 | 47.43 | 56.20 | 10.37 | 17.97 | 23.13 | 36.01 | 55.08 |
| $\overline{AVG}$ | CLIP | 14.48 | 21.43 | 14.33 | 26.95 | 25.39 | 22.59 | 22.11 | 16.63 | 21.14 | 13.05 | 14.92 | 3.82 | 5.09 | 10.65 | 10.97 | 16.36 |
| | FARE | 26.48 | 31.78 | 18.83 | 55.65 | 45.80 | 45.51 | 44.87 | 26.43 | 28.36 | 16.58 | 24.18 | 3.43 | 6.17 | 9.52 | 19.55 | 27.40 |
| | TeCoA | 36.56 | 35.95 | 19.54 | 62.80 | 59.87 | 51.71 | 51.67 | 28.46 | 27.41 | 14.79 | 28.82 | 3.52 | 5.84 | 15.11 | 20.40 | 30.42 |
| | PMG-AFT | 35.37 | 39.39 | 21.58 | 64.23 | 61.99 | 52.84 | 53.36 | 30.53 | 32.25 | 18.73 | 29.76 | 3.80 | 6.44 | 14.77 | 21.17 | 32.20 |
| | TGA-ZSR | 31.25 | 28.77 | 15.25 | 43.26 | 40.11 | 36.45 | 39.16 | 21.75 | 25.95 | 14.20 | 22.66 | 3.34 | 5.29 | 11.74 | 15.59 | 23.11 |
| | AdvFLYP | 27.74 | 43.18 | 23.05 | 64.93 | 62.36 | 53.09 | 50.39 | 32.55 | 35.02 | 23.26 | 31.00 | 4.36 | 7.57 | 11.72 | 22.45 | 33.21 |

Table 2: Recognition accuracy under different attack scenarios ($\epsilon = 1/255, 4/255$ and clean images) on downstream datasets of CLIP finetuned with AdvFLYP on `small-LAION` of 1M training images, compared with existing methods finetuned on ImageNet of roughly 1.2M training images.

TeCoA and PMG-AFT achieve higher robustness. However, this is at the cost of extreme clean accuracy decline (40.36 and 42.19, respectively, versus 53.28 for AdvFLYP). Another interesting finding is that AFT baselines finetuned on TinyImageNet leads the model to generalize better on downstream datasets of similar distribution, such as CIFAR10, CIFAR100 and STL10, which are also low-resolution general object classification datasets, under all evaluation settings.

**AdvFLYP *v.s.* AFT methods finetuned on ImageNet.** As can be seen from Table 2, AFT methods finetuned on ImageNet incur lesser loss of generalization on clean images, compared to when finetuned on TinyImageNet. This is due to the larger dataset size and better image quality of ImageNet, which effectively alleviates overfitting. Nonetheless, all baselines finetuned on ImageNet still exhibit signs of overfitting, with higher test accuracy on ImageNet, especially TGA-ZSR. Additionally, employing ImageNet as a proxy dataset for AFT benefits certain datasets that share more similar classes than others. For example, AFT baselines finetuned on ImageNet, which include a considerable amount of animal classes, transfer better to OxfordPets (Parkhi et al., 2012) and ImageNet-like general classification datasets. Among AFT baselines, the unsupervised method FARE achieves the best clean accuracy. However, FARE exhibits noticeably lower robustness levels, compared to supervised AFT. TGA-ZSR is shown to largely overfit to both the data distribution of ImageNet and the attack type during AFT. Our AdvFLYP performs consistently better than supervised methods in terms of generalization on clean images and different adversarial attack scenarios.

For each downstream dataset, we average the accuracy of a model under all scenarios including PGD-10 ($\epsilon = 1/255$), PGD-10 ($\epsilon = 4/255$), AutoAttack ($\epsilon = 1/255$) and clean images, for a comprehensive evaluation. When finetuned on a tiny portion of web-scale image-text data (`tiny-LAION`), our AdvFLYP achieves best overall results on 11 out of 14 downstream datasets, with an improvement of 4.86 points (19.1%). Additionally, AdvFLYP finetuned on `small-LAION` performs best on 12 out of 14 datasets, with an improvement of 1.01 points (3.1%). Results show that our AdvFLYP paradigm steadily enhances zero-shot adversarial robustness of CLIP across datasets of various domains, without overfitting to the distribution of any dataset, proving the importance of following the pre-training data and training objective of CLIP in AFT.

### 4.4 ANALYSIS ON ADVFLYP

This work proposes an AFT paradigm that practically resumes the pretraining process of CLIP, except that the training data are swapped for adversarial images. However, the behaviour of contrastive learning in a context of adversarial finetuning is understudied. This section explores other training settings of the AdvFLYP paradigm to provide insights into its working.

**Ablation studies.** On `tiny-LAION`, we ablate the default AdvFLYP to reveal contributions of each term of the formulated loss (Eq. 10) to its effectiveness. From Table 3, it can be seen that the contrastive loss $\mathcal{L}_{CLIP}$ as employed in CLIP's pretraining significantly improves its *zero-shot adversarial robustness* from the original CLIP's 3.04 to 33.06, which plays a major role in AdvFLYP. Adding logit-level regularization $\mathcal{L}_{logit}$ further im-

| (%) | Rob. Acc. | Clean Acc. | Avg. |
|---|---|---|---|
| $\mathcal{L}_{CLIP}$ | 30.41 | 53.02 | 41.72 |
| $\mathcal{L}_{CLIP} + \mathcal{L}_{logit}$ | **33.44** | 52.64 | 43.04 |
| $\mathcal{L}_{CLIP} + \mathcal{L}_{feat}$ | 30.74 | **55.11** | 42.93 |
| AdvFLYP | 33.06 | 53.28 | **43.17** |

Table 3: Ablation of training objective in AdvFLYP. The reported robust accuracy is tested under PGD-10 ($\epsilon = 1/255$). We report average accuracy over 14 downstream datasets.

proves transferability of robustness on downstream data, which is in line with the findings of PMG-AFT (Wang et al., 2024). In this work, we find that feature-level regularization $\mathcal{L}_{feat}$ is highly effectively in retaining generalization on downstream clean images. We introduce logit- and feature-level regularization into our AdvFLIP to reach a sweet spot between robustness and clean accuracy without further tuning their respective weights.

**Amount of training data.** Results in Table 1 and Table 2 show that a larger size of webscale data for AFT benefits both robustness and accuracy. We investigate the impact of training data amount on AdvFLYP in Fig. 2 (*left*). Interestingly, when the training data amount is scarce, both robustness and accuracy are data-limited and therefore, are not in conflict. Increasing the amount of collected web-scale data for AdvFLYP improves both robustness and accuracy considerably. When there are sufficient image-text pairs, the improvement plateaus and does not noticeably benefit from more training data.

**Batch size.** Different from existing AFT methods, which finetune $f_\theta$ through a cross-entropy *w.r.t.* the true labels of images, AdvFLYP employs the same contrastive loss in AFT as in the pretraining

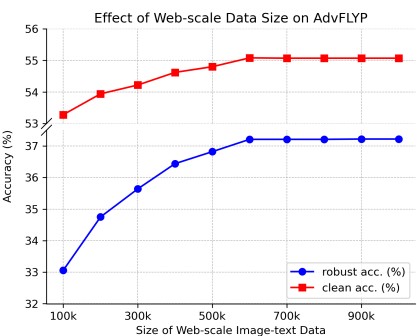 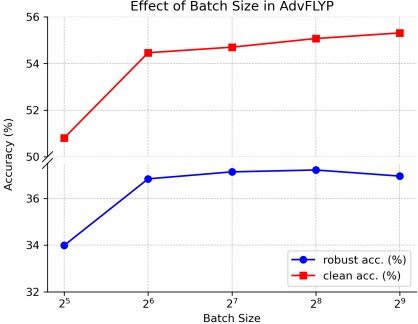

Figure 2: Adversarial robustness (PGD-10, $\epsilon = 1/255$) and clean accuracy of AdvFLYP paradigm under various training settings, averaged on 14 downstream datasets.

phase. As suggested in (Radford et al., 2021), a larger batch size is beneficial for contrastive learning, because it provides more negative samples in a single batch. In this experiment, we vary the batch size from 32 ($2^5$) to 512 ($2^9$) for AdvFLYP and report its performance in Fig. 2 (*right*). We adjust the number of epochs for each batch size to ensure a similar number of updates to model weights. We show that increasing the batch size for AdvFLYP improves clean accuracy significantly, which is in line with the findings of CLIP (Radford et al., 2021). In comparison, enlarging the batch size boosts the robustness at first. However, when the batch size increases to 512, robustness slightly declines, with additional gains of clean accuracy. This shows that both robustness and accuracy benefit from a sufficiently large batch size in contrastive learning in an AFT context. Further enlarging the batch size would trade robustness off for accuracy. We experiment with the maximum batch size of 512 due to hardware constraints.

**Unfreeze other components.** Existing AFT methods invariably finetune the vision encoder $f_\theta$ of CLIP, while keeping the text encoder $g_\phi$ frozen. Intuitively, resuming the pretraining recipe for AFT would result in whole finetuning of CLIP. In this experiment, we unfreeze more modules of CLIP to investigate the impact on AdvFLYP, and report the results in Table 4. It can be seen that unfreezing

| (%) | Rob. Acc. | Clean Acc. | Avg. |
|---|---|---|---|
| $f_\theta, g_\phi$ | 36.04 | 55.04 | 45.54 |
| $f_\theta, g_\phi$, others | 35.78 | 54.95 | 45.36 |
| $f_\theta\ (\mathcal{L}_{CLIP} \to \mathcal{L}_{I2T})$ | 36.29 | **55.30** | 45.80 |
| $f_\theta\ (\mathcal{L}_{CLIP} \to \mathcal{L}_{T2I})$ | 36.32 | 54.36 | 45.34 |
| $f_\theta$ | **36.82** | 54.80 | **45.81** |

Table 4: Impact of trainable CLIP modules on AdvFLYP finetuned on 500k web data.

more modules of CLIP in AdvFLYP does not lead to better robustness, indicating that $f_\theta$ is still the major component in adversarially robust CLIP. Unfreezing $g_\phi$ and more modules such as layer-wise normalization leads to slightly better clean accuracy. We also find that employing only the image-to-text loss, *i.e.*, the first term of $\mathcal{L}_{CLIP}$ (Eq. 1), leads to best clean accuracy. Utilizing the full contrastive loss $\mathcal{L}_{CLIP}$ and finetuning only $f_\theta$ achieves the best overall performance for AdvFLYP.

## 5 CONCLUSION

In this work, we propose a simple yet paradigm for adversarial finetuning, *Adversarially Finetune Like You Pretrain* (AdvFLYP), which practically resumes the training of CLIP with adversarial images, while retaining the training recipe as much as possible. This paradigm addresses the limitations of existing AFT methods, which invariably finetune the CLIP model on adversarial images on a proxy classification dataset through a cross-entropy loss, compromising the generalization of CLIP. Our AdvFLYP paradigm employs web-scale image-text data that follows the distribution of CLIP's pretraining data, and finetunes the same contrastive loss as employed during CLIP's pretraining. We additionally find that logit- and feature-level regularization benefits robustness and clean accuracy, respectively. AdvFLYP outperforms existing AFT methods commonly finetuned on TinyImageNet and ImageNet by 19.1% and 3.1%, respectively, while alleviating overfitting to any proxy dataset distribution. We analyse the behaviour of AdvFLYP and show that a sufficient large training data size and training-time batch size are crucial to both downstream robustness and accuracy, throwing light on the behaviour of contrastive learning in an adversarial finetuning context of CLIP.

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
