# OpenReview forum: "AdvFLYP: Adversarially Finetune Like You Pretrain for Zero-shot Robustness of CLIP"
_ICLR.cc/2026/Conference — ICLR 2026 Conference Withdrawn Submission_

### Official Review · Reviewer_9TTH · 2025-10-30

**Soundness:** 2
**Presentation:** 2
**Contribution:** 2
**Rating:** 4
**Confidence:** 3

**Summary:**

This paper proposes AdvFLYP (Adversarially Finetune Like You Pretrain), a new adversarial finetuning paradigm for CLIP that retains the training recipe of its pretraining stage. Instead of using proxy classification datasets and cross-entropy losses as in existing AFT methods, AdvFLYP performs adversarial finetuning with web-scale image-text pairs and the same contrastive loss used in CLIP pretraining, combined with logit-level and feature-level regularization. Extensive experiments on 14 downstream datasets demonstrate that AdvFLYP achieves superior zero-shot robustness while better preserving generalization on clean images compared with prior AFT approaches.

**Strengths:**

- A meaningful attempt to update the adversarial training paradigm by aligning it more closely with the CLIP pretraining recipe.
- Demonstrates consistent improvements on both clean accuracy and robustness accuracy across multiple datasets and attack settings.

**Weaknesses:**

- The paper aims to address the weak generalization issue of existing adversarial training methods by designing a framework that improves generalization. Interestingly, AdvFLYP not only enhances generalization but also achieves stronger robustness. However, the paper does not provide a clear explanation for why AdvFLYP yields higher robustness than other methods, which limits the depth of the work.
- The proposed paradigm relies on web-scale image–text data that may contain semantic noise, in contrast to the cleaner proxy datasets used in previous works. This could be a crucial factor affecting both generalization and robustness, yet the paper does not discuss this aspect.
- The role of L_feat is questionable. In [1], the logit-level regularization terms (L_general and L_clean) already include weighting parameters α and β, and when β = 0, they also demonstrate the ability to retain generalization on downstream clean images (the same effect that L_feat aims to achieve in this paper). Therefore, introducing L_feat may not be necessary.

[1] Pre-trained Model Guided Fine-tuning for Zero-shot Adversarial Robustness. CVPR 2024

**Questions:**

Please refer to the weaknesses.

---

### Official Review · Reviewer_iUEE · 2025-10-31

**Soundness:** 2
**Presentation:** 3
**Contribution:** 3
**Rating:** 2
**Confidence:** 5

**Summary:**

This paper proposes a new adversarial finetuning method, AdvFLYP, to improve zero-shot robustness of the CLIP model on downstream datasets that are different from the finetuning dataset. The loss consists of three terms: the same contrastive loss as in CLIP’s pretraining, a logit-level regularization, and feature-level regularization. AdvFLYP outperforms existing AFT methods commonly finetuned on TinyImageNet and ImageNet by 19.1% and 3.1%.

**Strengths:**

The motivation of this paper is clear, and the problem it aims to address is meaningful, as pretrained models are commonly used without any finetuning for some downstream tasks.

**Weaknesses:**

1.	Unfair comparison settings. Since the baseline methods are finetuned on tiny-imagenet and imagenet in Tables 1 and 2, the authors should also follow these settings. Otherwise, it is not clear whether the improvement on downstream datasets is benefited by the proposed method or the choice of the finetuning dataset. Additionally, it would be beneficial to include an ablation study of the finetune dataset.

2.	Epsilon equals 1/255 is not the most common setting. The author should report the results of the auto attack with epsilon equal to 2/255 and 4/255. However, these results are absent in the paper.

**Questions:**

1.	The definition of the loss term (Equation 9, 10) is a bit strange, as it encourages the current model to have a similar feature to the initial model on adversarial examples. As the initial model is vulnerable to adversarial examples, this feature should be meaningless in principle. Why not encourage it to have a similar feature on clean examples?
2.	L logit and L feat seem abundant, according to the ablation study (Table 3).  It appears that using either one or both is simply a matter of balancing robust accuracy against clean accuracy. What if you use a TRADE-like loss?

---

### Official Review · Reviewer_DDgX · 2025-10-31

**Soundness:** 3
**Presentation:** 3
**Contribution:** 3
**Rating:** 4
**Confidence:** 4

**Summary:**

The paper introduces a novel approach called Adversarially Finetune Like You Pretrain (AdvFLYP) to improve the adversarial robustness of CLIP, a vision-language model. Existing adversarial finetuning (AFT) methods often compromise the model's generalization by using mismatched training data and objectives. AdvFLYP addresses this by retaining the pretraining structure of CLIP, finetuning it with adversarial images aligned to their respective textual descriptions using a contrastive loss. The method outperforms traditional AFT techniques like those finetuned on TinyImageNet and ImageNet, while maintaining strong generalization on clean images. The study also emphasizes the importance of large training datasets and batch sizes for optimal robustness and accuracy, providing insights into contrastive learning within adversarial finetuning contexts.

**Strengths:**

（1）Well-Motivated Paradigm
The paper introduces a clear idea—adversarially finetuning CLIP with the same recipe as pretraining—which effectively bridges the gap between pretraining and robustness adaptation.

（2）Comprehensive Experimental Validation
The authors conduct extensive experiments on 14 downstream datasets, providing strong empirical evidence that AdvFLYP consistently outperforms prior AFT methods in both robustness and generalization.

**Weaknesses:**

（1）Limited Novelty

The paper’s main contribution—the use of contrastive loss during adversarial finetuning—appears to be an incremental extension of the established Finetune Like You Pretrain (FLYP) paradigm into the adversarial setting. Since adversarial finetuning is essentially a variant of finetuning, the methodological innovation of AdvFLYP is relatively modest.

（2）Potentially Unfair Comparison

When demonstrating the advantage of contrastive learning, the comparisons should ideally be performed on the same dataset. However, the paper compares competing AFT methods (TeCoA, PMG-AFT, TGA-ZSR, etc.) finetuned on ImageNet with AdvFLYP trained on web-scale LAION. Although the dataset sizes are matched, the authors themselves acknowledge that differences in data distribution can have a substantial impact on performance, which raises concerns about the fairness of the comparison.

（3）Insufficient Ablation Analysis

The ablation study does not fully disentangle the effects of the two key factors—(a) adopting the contrastive learning objective and (b) using data that follows CLIP’s pretraining distribution. The authors combine both changes in their experiments, making it unclear which factor contributes more significantly to the observed performance gains. A more systematic analysis isolating these effects would strengthen the paper’s conclusions.

**Questions:**

（1）Request for Fair Comparison on the Same Dataset.

Please provide comparative results where all methods are trained and evaluated on the same dataset. This would more clearly isolate the advantage of the proposed contrastive learning objective, as current comparisons involve different datasets (e.g., TinyImageNet vs. LAION subsets), which may introduce confounding factors related to data distribution differences rather than methodological improvements.

（2）Request for Additional Ablation Studies.

Please include additional ablation experiments to disentangle and analyze the individual contributions of (a) adopting the contrastive learning objective and (b) using data that follows CLIP’s pretraining distribution. Clarifying the relationship and relative importance of these two components would provide stronger empirical evidence for the core claims of AdvFLYP and better justify its design choices.

（3）Could the authors explain why the benefit of the contrastive learning strategy appears to diminish as the dataset size increases? In the reported results, the performance gap between AdvFLYP and baseline methods becomes smaller when training on larger datasets (e.g., ImageNet-scale). It would be valuable to understand whether this trend is due to reduced overfitting on larger datasets, saturation effects in contrastive learning, or other factors related to data diversity or optimization dynamics.

---

### Official Review · Reviewer_aLm8 · 2025-11-01

**Soundness:** 3
**Presentation:** 3
**Contribution:** 2
**Rating:** 4
**Confidence:** 4

**Summary:**

This paper proposes AdvFLYP, a new AFT paradigm that "Finetunes Like You Pretrain" by resuming the training on web-scale image-text pairs and using the original contrastive loss. Experiments show this approach significantly improves the robustness-generalization trade-off, outperforming baselines trained on equivalent-sized datasets like TinyImageNet and ImageNet by 19.1% and 3.1%, respectively, across 14 downstream datasets. The work demonstrates that aligning the AFT recipe with the pretraining recipe is crucial for maintaining zero-shot capabilities.

**Strengths:**

1. Simple and Intuitive Hypothesis: The paper's central premise that AFT should align with the model's pretraining recipe is logical. The authors provide clear evidence for this hypothesis in their preliminary experiments, which demonstrate that standard AFT methods overfit to their proxy datasets. This clean motivation sets the stage perfectly for the proposed solution.

2. Strong and Comprehensive Empirical Results: The paper demonstrates a clear and significant improvement over existing SOTA AFT methods. The comparison is fair, as AdvFLYP is compared against baselines using equivalent-sized training datasets (100k tiny-LAION vs. 100k TinyImageNet, and 1M small-LAION vs. 1.2M ImageNet ). Achieving a 19.1% (4.86 point) average improvement in the 100k-scale comparison is substantial. The method consistently achieves a superior balance between robust accuracy and clean accuracy across 14 diverse datasets (Tables 1 & 2).

**Weaknesses:**

1. Incremental Insights: The paper's key analyses (Section 4.4) on the importance of data scale and batch size are not new insights. The original CLIP paper (Radford et al., 2021) already established that contrastive learning thrives on massive data and very large batch sizes. This paper's finding that adversarial contrastive learning also benefits from these factors is an incremental extension, not a fundamental discovery.

2. Computational Cost of the Attack Objective: The inner-loop maximization (Eq. 4) generates a batch-wise perturbation by maximizing the $\mathcal{L}_{CLIP}$ loss. This objective requires computing an $N \times N$ similarity matrix (where N is the batch size) and its corresponding loss on every step of the PGD algorithm. For practical adoption, a clear analysis of the training time and computational overhead compared to baselines is essential.

3. Clarification needed for the "Pretraining Recipe" Scale: The paper's title and premise ("Finetune Like You Pretrain") evoke the scale of CLIP's original pretraining (400M pairs, 32k batch size). However, the experiments use 1M pairs and a max batch size of 512. More importantly, the analysis in Fig. 2 shows performance plateauing around 700k data samples and a batch size of 256-512 (for robustness). This finding, while interesting, seems to contradict the core premise that matching the pretraining recipe is the goal in terms of the amount of training data. If performance saturates this early, it suggests that the "pretraining recipe" is more about the style (web-data, contrastive loss) than the scale. This discrepancy should be discussed. It's an opportunity to refine the paper's conclusion: perhaps full-scale data is unnecessary for robust finetuning.

**Questions:**

1. Could you please quantify the training-time overhead of using the batch-wise contrastive loss for perturbation generation, compared to a baseline using a standard cross-entropy loss?
2. In your ablation (Table 4), you found that finetuning only the vision encoder $f_{\theta}$ gave the best overall performance, and unfreezing the text encoder $g_{\phi}$ did not help. Could you give any insights or clarification on this?
3. How sensitive is the final robustness to the number of PGD steps used during training? You used K=2, but is it possible that the complex contrastive objective requires more steps to find strong attacks, and that the model is "falsely" robust to a weak training-time attack?

---

### Note · Authors · 2025-11-14

**Comment:**

We thank the reviewers and the AC for their valuable time and efforts. Given the current status, we decide to withdraw this submission. We will keep on improving our work based on your constructive feedback.

Best regards,

#5221 Authors

**Withdrawal Confirmation:**

I have read and agree with the venue's withdrawal policy on behalf of myself and my co-authors.